# Inactivation of the UL37 Deamidase Enhances Virus Replication and Spread of the HSV-1(VC2) Oncolytic Vaccine Strain and Secretion of GM-CSF

**DOI:** 10.3390/v15020367

**Published:** 2023-01-27

**Authors:** Carolyn M. Clark, Nithya Jambunathan, Therese M. A. Collantes, Konstantin G. Kousoulas

**Affiliations:** 1Division of Biotechnology and Molecular Medicine, Department of Pathobiological Sciences, School of Veterinary Medicine, Louisiana State University, Baton Rouge, LA 70803, USA; 2School of Medicine, Indiana University, Indianapolis, ID 46202, USA; 3College of Veterinary Medicine, University of the Philippines Los Baños, Los Baños 4031, Laguna, Philippines

**Keywords:** herpesvirus, VC2, UL37, GM-CSF, glycoprotein K, UL20, AlphaFold, functional modeling

## Abstract

The HSV-1 (VC2) live-attenuated vaccine strain was engineered with specific deletions in the amino termini of glycoprotein K (gK) and membrane protein UL20, rendering the virus unable to enter neurons and establish latency. VC2 replicates efficiently in epithelial cell culture but produces lower viral titers and smaller viral plaques than its parental HSV-1 (F) wild-type virus. VC2 is an effective live-attenuated vaccine against HSV-1 and HSV-2 infections in mice and guinea pigs and an anti-tumor immunotherapeutic and oncolytic virus against melanoma and breast cancer in mouse models. Previously, we reported that the gK/UL20 complex interacts with the UL37 tegument protein, and this interaction is essential for virion intracellular envelopment and egress. To investigate the potential role of the UL37 deamidase functions, the recombinant virus FC819S and VC2C819S were constructed with a C819S substitution to inactivate the UL37 predicted deamidase active site on an HSV-1(F) and HSV-1(VC2) genetic background, respectively. FC819S replicated to similar levels with HSV-1(F) and produced similar size viral plaques. In contrast, VC2C819S replication was enhanced, and viral plaques increased in size, approaching those of the wild-type HSV-1(F) virus. FC819S infection of cell cultures caused enhanced GM-CSF secretion in comparison to HSV-1(F) across several cell lines, including HEp2 cells and cancer cell lines, DU145 (prostate) and Panc 04.03 (pancreas), and primary mouse peritoneal cells. VC2 infection of these cell lines caused GM-CSF secretion at similar levels to FC819S infection. However, the VC2C819S virus did not exhibit any further enhancement of GM-CSF secretion compared to the VC2 virus. These results suggest that the UL37 deamidation functions in conjunction with the gK/UL20 complex to facilitate virus replication and GM-CSF secretion.

## 1. Introduction

### 1.1. Clinical Disease

Herpes simplex virus type 1 (HSV-1) is an ancient human pathogen of the Alphaherpesvirinae subfamily that evolved alongside the first humans [1]. Typical infection begins in the epithelial cells around the lips. The virus travels and spreads through the axonal termini of sensory neurons to the neuronal cell body in the trigeminal ganglia (TG). In the TG, the virus establishes a period of latency. During latency, viral gene transcription is suppressed almost completely [2]. However, the virus reactivates during immune suppression, initiating replication and spreading back to peripheral tissues. The most common disease manifestation is blistering around the lips, called “cold sores.” However, HSV-1 is associated with more severe disease when the virus reacts to ocular surfaces causing herpetic keratitis or when the virus reacts to the central nervous systems (CNS), resulting in encephalitis and meningitis. Moreover, HSV-1 can infect genital tissues, where the virus establishes latency in the dorsal root ganglia and reactivation results in herpes genitalia. For these reasons, developing an efficacious vaccine is of the utmost importance to ameliorate the disease caused by HSV-1 infection. 

### 1.2. Viral Entry

HSV-1 has a wide range of tissue tropism and can enter cells via fusion of the viral envelope with cellular plasma membranes or endocytosis [3,4]. Fusion of the viral envelope with cellular plasma membranes is exclusively utilized in neuronal axonal entry and results in capsid and tegument proteins deposited directly into the cytoplasm [5,6]. Consequently, the establishment of latency in neurons is dependent on viral fusion. However, HSV-1 can also enter epithelial cells via pH-dependent and -independent clathrin-mediated endocytosis [7,8]. In this case, the virus enters the cell in an early endocytic vesicle. The virus fuses with the endosome to deposit the capsid and tegument proteins into the cytoplasm. These two routes of entry, fusion, and endocytosis, contribute to the extensive tissue tropism of HSV-1 as it can enter a wide range of cell types.

The HSV-1(VC2) strain has specific deletions in the amino termini of glycoprotein K (gK) and the membrane protein UL20. These deletions alter gB-mediated fusion resulting in VC2 being unable to enter cells, including neuronal axons, via fusion of the viral envelope with plasma membranes. However, VC2 can enter epithelial and fibroblast cells in cell culture via endocytosis and replicate efficiently [9,10]. Intramuscular immunization of mice and guinea pigs with VC2 generates protective ocular immune responses against the virulent HSV-1(McKrae) human ocular strain in mice [11]. Moreover, VC2 intramuscular immunization of mice and guinea pigs generates protective immune responses against both virulent HSV-1 and HSV-2 infections of genital tissues [11,12,13,14]. Additionally, VC2 has been utilized as a vector to express malaria and influenza genes to protect against lethal challenges with those pathogens [15,16]. These results suggest that VC2 induces an adjuvant effect.

### 1.3. The UL37 Tegument Protein

HSV-1 structure includes many viral proteins embedded in the viral envelope, several capsid proteins surrounding the genome, and a layer of tegument proteins between the envelope and capsid. These tegument proteins serve many essential roles in the virus life cycle, such as facilitating virion transport, initiating viral gene transcription, virus assembly, and egress. While many of their essential functions are well described, these proteins often include multiple distinct functional domains with separate roles. The HSV-1 UL37 tegument protein is 1123aa long and is highly conserved among all members of the neurotropic Alphaherpesvirinae subfamily. UL37 is a deamidase involved in immunomodulation [17,18,19,20,21,22,23]. Specifically, the UL37 catalytic site at C819 acts to deamidate multiple cellular pathogen recognition receptors (PRRs), cGAS, and RIG-I [21,22,23]. Deamidation of RIG-I and cGAS during wild-type HSV-1 infection blocks their signaling function and reduces the expression of type-I interferons. When the C819 site is interrupted with the C819S amino acid substitution, RIG-I and cGAS function is maintained, and expression of type-I interferons increases [21,23].

The UL37 protein is required for cytoplasmic virion envelopment and can be transported to the cis-phase of the Golgi apparatus as a complex with the UL36 tegument protein in the absence of capsid formation [24]. The UL37 protein physically interacts with the gK/UL20 heterodimer, and this interaction is essential for cytoplasmic virion envelopment, infectious virus production, and cellular egress [25].

### 1.4. HSV-1 and Cytokine Regulation

HSV-1 has developed a highly complex and multi-faceted approach to evading, suppressing, and modulating the immune response. A wide range of pathogen recognition receptors (PRRs) recognizes HSV-1, including TLR2, TLR3, TLR4, TLR9, RIG-I, MDA5, and cGAS (reviewed in [26]). Additionally, many viral proteins can suppress PRRs and their downstream adaptors, including US3, UL11, UL37, ICP0, and VP16 (reviewed in [26]). These PRRs cause increased expression of inflammatory cytokines, including TNF, CXCL10, interferon β (IFN-β), and many other interferon-stimulated genes (ISGs).

Our laboratory has shown that VC2 has exceptional promise as an oncolytic and immunotherapeutic for the treatment of melanoma and other cancers; specifically, we have reported that VC2 is efficacious in the treatment of melanoma in a mouse immunocompetent and syngeneic model system and can generate strong anti-tumor immune responses [27,28]. This strong VC2 adjuvant effect may be due to the upregulation of innate immune responses. Currently, three HSV-1-based oncolytic and immunotherapeutic strains are licensed for human use. Talimogene Laherparepvec (T-VEC; Imlygic™, Amgen Inc., Thousand Oaks, CA, USA) is an HSV-1 strain with deletions of both copies of ICP 34.5 and an insertion of GM-CSF into each location. The GM-CSF insertion improves local and systemic anti-tumor immunity [29]. T-VEC has been licensed for human anti-melanoma use and is currently tested against other cancers. G207 (Treovir, Inc., Bala Cynwyd, PA, USA) is an HSV-1 (F) background with a deletion of both ICP 34.5 genes and an insertion of lacZ to disable UL39, which shows strong efficacy as a therapy for malignant glioma [30,31]. Delytact (G47Δ; teserpaturev; Daiichi Sankyo, Co, Tokyo, Japan) has been recently conditionally approved in Japan. It is the most recent version of G207, but with an additional deletion of ICP47 [32,33]. These viruses demonstrate the need for more efficacious herpes oncolytic virotherapy options to treat various cancers.

One of the major cytokines in the recruitment of antigen-presenting cells, specifically dendritic cells, is granulocyte-macrophage colony-stimulating factor (GM-CSF). GM-CSF expression by Talimogene laherparepvec (T-VEC; Imlygic™, Amgen Inc.) stimulates anti-tumor immune responses. GM-CSF is 127aa in length with many glycosylation sites and is secreted by many cell types, including lymphocytes, fibroblasts, and endothelial cells [34] (reviewed in [35]). GM-CSF is vital in the proliferation and differentiation of many myeloid and granulocytic cells [36]. 

Herein, we show that the C819S mutation increases the VC2 virus replication and spread, indicating that the UL37 deamidase function is intimately involved in the role of the gK/UL20 protein complex in virus replication and spread. We show that VC2 infection enhances the secretion of GM-CSF from several cancer cell lines and primary myeloid cells. Inactivation of the UL37 deamidation active specified by the wild-type parental virus HSV-1(F), but not VC2, results in enhanced GM-CSF secretion. These data indicate that this UL37 deamidation-dependent GM-CSF secretion is functionally associated with the UL37/gK/UL20 interactome.

## 2. Materials and Methods

### 2.1. Cell Lines

Vero (African green monkey kidney cells), HEp-2 (human laryngeal cancer/HELA contaminant), Panc 04.03 (human pancreatic cancer), and DU 145 (human prostatic cancer) were obtained from the American Type-Culture Collection (ATCC) (Rockville, MD, USA). L929-/-cGAS (in this paper called “L929KO”) and L929-/-cGAS reconstituted with human cGAS (called “L929R”) were a gift from Pinghui Feng; L929 cells are mouse subcutaneous fibroblasts. Veros, HEp-2, L929KO, and L929R, were maintained in Dulbecco’s Modified Eagle medium (Gibco-BRL, Grand Island, NY, USA) supplemented with 10% fetal bovine serum (Gibco-BRL, Grand Island, NY, USA) and 100 ug/mL Primocin (Invitrogen, INC., Carlsbad, CA, USA). DU 145 cells were maintained on Minimum Essential medium supplemented with 10% fetal bovine serum (Gibco-BRL, Grand Island, NY, USA) and 100 ug/mL Primocin (Invitrogen, INC., Carlsbad, CA, USA). Panc 04.03 were maintained on RPMI-1640 complete medium (Gibco-BRL, Grand Island, NY, USA) supplemented with 20 U/mL recombinant human insulin, 15% fetal bovine serum (Gibco-BRL, Grand Island, NY, USA), and 100 ug/mL Primocin (Invitrogen, INC., Carlsbad, CA, USA).

### 2.2. Mice

Peritoneal cells were harvested from female Balb/CJ mice (8–10) weeks old. Mice were purchased from Jackson Laboratories (Bar Harbor, ME, USA) and housed at the Louisiana State University School of Veterinary Medicine laboratory animal facility.

### 2.3. Viruses

The wild-type virus used in this study was HSV-1 (F). HSV-1 (VC2) was previously generated in our lab on an HSV-1 (F) background to include gK Δ31–68 and UL20 Δ4–22 [9]. Previously, the C819S amino acid substitution was described as inactivating the deamidase activity of UL37 [22]. We added the UL37 C819S substitution to a WT HSV-1 (F) virus and HSV-1 (VC2) (Figure 1). The VC2 and C819S virus we called VC2C819S. Virus stocks were grown on Vero cells, and titer was calculated using a methylcellulose plaque assay stained with crystal violet.

FC819S and VC2C819S were developed using two-step Red recombination mutagenesis and synthetic oligonucleotides in Escherichia coli. The forward primer was 5′-GGGGCCCTGG CCCCCCGAGGCCATGGGGGACGCGGTGAGTCAGTACAGCAGCATGTATCACGAC GCCAAGCGCGCGCTGGTCGCGTCCCTAGGATGACGACGATAAGTAGGG-3′, and the reverse was 5′-GTGCGCCGTGGTTTCGGTGATGACGGAACGCAGGCTCGCG AGGGACGCGACCAGCGCGCGCTTGGCGTCGTGATACATGCTGCTGTACTGACTCACCGCGTCCCCCATGGCCTCGGGGGGCAACCAATTAACCAATTCTGATTAG-3′. These oligonucleotides were implemented on the bacterial artificial chromosome (BAC) plasmid pYEbac102-VC2 carrying HSV-1 (VC2) previously developed in our lab. The sequence was confirmed using viral genome sequencing.

### 2.4. Replication Kinetics

Growth kinetics were performed on confluent 12 well plates of Hep2, L929R, and L929KO in triplicate. Viruses were infected at an MOI of 1.0 and then adsorbed by rocking for 1 h at 4 °C. Plates were then rocked at RT for 1 h. Then plates were moved to incubate at 37 °C for 24 h post-infection (hpi). For whole-cell titers, plates were frozen at −80 °C and thawed three times, and cell lysates were collected at that time point. For secreted titers, cell media was collected and frozen at −80 °C. Fresh media was added to the plates. Then, plates were frozen and thawed three times to collect cell lysates. All samples were titrated on Vero cells and stained with crystal violet. The average and standard error of the mean for each sample was calculated for each virus and time point.

### 2.5. ELISA

The levels of secreted cytokines were determined by ELISA. Confluent monolayers were infected at an MOI of 5, and conditioned media was collected at 12 hpi. These samples were analyzed for secreted GM-CSF using the GMCSF human ELISA kit and following the manufacturer’s guidelines (Invitrogen, INC., Carlsbad, CA, USA).

### 2.6. Predicted Modeling of UL37 C819S and gK/UL20 Complex

To compare differences in the structure of WT UL37 and UL37 C819S, we generated tertiary structural models using AlphaFold2 (Google Colaboratory). UL37 WT and UL37 C819S were aligned using PyMol to determine the root mean standard deviation value (RMSD). To determine structural changes and differences in membrane interaction of the gK/UL20 complex, we predicted protein–protein complex formation using AlphaFold2- multimer (Google Colaboratory). The amino acid sequences of human alphaherpesvirus-1 glycoprotein K (gK) (Accession Number AFH41179.1) and UL20 (Accession Number QFQ61390.1) were retrieved from GenBank and used as input, alongside the gKΔ31–68/UL20Δ4–22 amino acid sequences. The predicted models were inserted into a bilipid membrane using MemProtMD. Briefly, MEMEMBED was used to orient the protein relative to a lipid membrane. A box of 80 Å in size (*z*-axis) was generated to contain the protein. The protein was contained in the box, and the *x* and *y* axes were determined by providing a distance of 30 Å from the protein. Dipalmitoylphosphatidylcholine (DPPC) lipid models were used to insert the protein into a bilipid membrane. The bilipid membrane was built with the following using the MARTINI 2.1 forcefield: 1-palmitoyl-2-oleoyl-sn-glycero-3-phosphoethanolamine (POPE), 1-palmitoyl-2-oleoyl-sn-glycero-3-phosphoglycerol (POPG), cardiolipin (CL) and glucosyl-lipopolysaccharide (UDP1). The coarse-grained molecular dynamics (CGMD) simulation was completed at 60 ns. At the end of the simulation, a snapshot was taken to convert the coarse-grained resolution to atomistic resolution using CG2AT-align.

### 2.7. Statistical Analyses

Experiments were in experimental triplicate unless indicated otherwise. Replication kinetics were performed in technical triplicate and averaged. ELISA data were performed in technical duplicate and averaged. Replication kinetics were log-transformed. Statistical analyses were performed using a two-way analysis of variance (ANOVA) or a Student’s *t*-test; *p* < 0.05 was considered significant. Tukey post-test adjustments were applied to multiple comparisons between each treatment group and the control. All analyses were performed using GraphPad (version 9) software (Graphpad Software, San Diego, CA, USA).

## 3. Results

### 3.1. Construction and Characterization of F-C819S and VC2C819S Viruses

The HSV-1 (VC2) strain has been derived from HSV-1(F) by engineering the gK Δ31–68 and UL20 Δ4–22 amino acid deletions [9]. To abrogate the deamidase function of the UL37 C819 site, we engineered a C819S amino acid substitution in both the HSV-1(F) and VC2 genetic backgrounds using double-Red recombination in conjunction with the cloned viral genomes as bacterial artificial chromosomes (bac), as previously described [21,22,23,37] (Figure 1). The engineered mutations and the absence of undesired mutations were confirmed by whole viral genome sequencing in our core facility GeneLab.

### 3.2. Viral Spread and Growth Kinetics

As previously reported, VC2 produces smaller non-syncytial plaques than F (Figure 2A) [9]. The plaque morphology of the FC819S virus was consistent with HSV-1(F) (Figure 2A). The C819S mutation engineered into VC2 increased the average plaque size of the virus rendering it similar to that of the wild-type HSV-1(F). In addition, these viral plaques exhibited weak syncytial morphology relative to VC2 (Figure 2A).

Next, we investigated the impact of gKΔ31–68, UL20Δ4–22, and UL37C819S mutations on viral replication. Viruses were grown in HEp-2 cells at an MOI of 1 for 0, 6, 12, 24, and 48 h post-infection. HSV-1(F) and FC819S viruses exhibited similar growth kinetics, while VC2 titers were approximately ten-fold lower at 24 hpi but approached wild-type titers at 48 hpi (Figure 2B). The C819S mutations rescued the slower VC2 replication kinetics producing similar titers at 24 hpi with the wild-type HSV-1 (F) (Figure 2B). Extracellular versus intracellular virion titers of the C819S mutations did not show any significant defect in virion egress for either FC819S or VC2C819S mutant viruses (Appendix A).

### 3.3. VC2 and FC819S Upregulate GM-CSF Protein Secretion

The UL37 C819 deamidates the pathogen recognition receptor cGAS. Using L929 cells without CGAS (L929KO) and those reconstituted with human cGAS (L929R), we tested the ability of different viruses to replicate in the presence and absence of cGAS. VC2 and VC2C819S replicated to a similar extent and less efficiently than F and FC819S in L929R cells, as seen in Vero and other cell lines. F, FC819S, and VC2 replicated to similar levels in L929KO and L929R cells. However, VC2C819S replicated significantly more efficiently than both VC2 and F viruses (Figure 3A,B). These results were further confirmed by determining viral replication kinetics at MOI of 1 and 0.1 (Appendix A).

VC2 produces a substantial adjuvant effect [11]. Therefore, we investigated the secretion of different cytokines and found a significant upregulation of GM-CSF secretion by infected cells (Figure 3C–F). Conditioned media was collected from infected HEp-2 (Figure 3C), primary mouse peritoneal cells (Figure 3D), DU145 (Figure 3E), and Panc 04.03 (Figure 3F) to quantify secreted GM-CSF protein by ELISA. FC819S and VC2 upregulated GM-CSF expression in all cell types relative to HSV-1(F), but VC2C819S infection did not cause upregulation of GM-CSF secretion in comparison to VC2.

### 3.4. Secondary Virion Envelopment in the Golgi

Viruses with deletions of the entire gK, UL20, or UL37 genes all exhibit similar phenotypes, with unenveloped capsids accumulating in the cytoplasm. All three proteins are involved in viral egress. We investigated the impact of the partial deletions of gKΔ31–68, UL20Δ4–22, and UL37C819S mutations on virus egress using TEM. At 18 h post-infection, cells infected with FC819S and VC2C819S appeared to have many virions attached to the extracellular plasma membrane in comparison to their respective parental strains HSV-(F) and VC2 (Figure 4).

### 3.5. Modeling Tertiary Structures of gK, UL20, and UL37

HSV-1 UL37 tertiary structure was modeled using AlphaFold2 as previously described [38]. The C819 location was identified in the center of an elongated alpha helix within the C-terminal half (Figure 5A). Modeling of UL37-C819S did not drastically change the overall conformation of the UL37 structure but opened the clip-like structure of UL37 into a potentially more accessible conformation (Figure 5B). The root mean standard deviation (RMSD) was calculated to be 7.582 A from the alignment of the wild-type UL37 to UL37C819S.

Because gK and UL20 are known to complex together, we used AlphaFold2-multimer to model the tertiary structure of the gK/UL20 complex for both wild-type gK/UL20 and ΔgK31–68/ΔUL204–22. The gK/UL20 complex is embedded in the viral envelope. Therefore, these protein complexes were modeled in association with bilipid membranes using MemProtMD (Figure 5C,D). The wild-type gK/UL20 complex is embedded in the model membrane exposing certain domains intracellularly with a large alpha helix of gK located on the extracellular side of the membrane and the amino-terminal beta sheets on the cytoplasmic side of the membrane (Figure 5C). However, the VC2 gK31–68/UL204–22 protein complex is predicted to be more embedded into the membrane preventing cytoplasmic domains from being fully exposed in comparison to the wild-type gK/UL20 protein complex (Figure 5D). This shift potentially limits access to the gK/UL20 intracellular binding sites by shifting the complex into the membrane while making the extracellular domains of gK/UL20 more exposed and accessible for protein–protein interactions with gB and other viral glycoproteins.

## 4. Discussion

Previous work in our laboratory has focused on identifying distinct functional domains and binding sites of gK and its binding partner, UL20. While best characterized for their roles in egress, they function in physical association with the UL37 tegument protein to facilitate cytoplasmic virion envelopment; they are also important modulators of viral entry and fusion through their interactions with the sole fusion glycoprotein gB [10,39]. Deleting the gK and UL20 binding sites with gB modifies viral entry preventing the virus from entering via the fusion of the viral envelope with cellular membranes, including neuronal axons [10]. This led to the development of VC2 as a vaccine and oncolytic strain [9,11,14,15,40,41]. Herein, we show that inactivation of the UL37 deamidase catalytic site restores the VC2 virus replication and spread defects indicating that this amino acid change alters the ability of the UL37/gK/UL20 complex to function in virus replication and spread. Importantly, we show for the first time that VC2 causes the enhanced secretion of GM-CSF, which may partially explain its efficacy in melanoma cancer treatment in mice [27,28], and this effect is also associated with the UL37/gK/UL20 interactome.

### 4.1. Construction and Characterization of FC819S and VC2C819S Viruses

We have shown previously that the gK/UL20 heterodimer interacts with the UL37 protein, and both function in cytoplasmic virion envelopment and egress [42]. Specifically, deleting any of these three proteins prevents secondary envelopment [43,44,45,46,47,48]. Recent reports have identified additional UL37 functional domains active at other steps in the viral life cycle [22,49]. Notably, the C819 catalytic site of UL37 deamidates and deactivates two cytosolic PRRs, cGAS, and RIG-I [21]. Based on the known interactions between gK/UL20 and UL37, we investigated whether the deamidation function of UL37 is functionally associated with gK/UL20.

The VC2 virus produces a smaller plaque and replicates less efficiently than its parental wild-type HSV-1(F) [9]. The FC819S mutant virus produces similar plaque size and replication kinetics to the HSV-1(F) virus. In contrast, the VC2C819S virus produces viral plaques that are much larger than its parental virus VC2 and similar in size to the HSV-1(F) prototypic virus. In addition, the VC2C819S viral plaques exhibited a weak syncytial phenotype. This result indicates that the gK/UL20 viral phenotype is intimately associated with the UL37 catalytic deamidation function. One potential explanation is that the deamidation site directly interacts with the gK/UL20 heterodimer or functions to deamidate a functional site on gK/UL20 proteins. The known targets of the UL37 C819 deamidase site are cGAS and RIG-I; however, the replication curves in cells with cGAS knocked out and reconstituted are very similar (Figure 3A,B). The VC2 virus exhibited lower replication kinetics in the L929KO compared to the wild-type virus, as well as compared to VC2 replication in the L929R cells. However, the VC2C819S virus exhibited enhanced replication in comparison to VC2, approaching higher titers than HSV-1(F). This suggests that cGAS is intimately associated with VC2 growth kinetics, most likely because it interacts with UL37. Alternatively, cGAS may act through other intracellular cGAS-dependent functions that can modulate the gK/UL20/UL37 complex. Computational modeling of the UL37 protein carrying the C819S mutation reveals that this amino acid change may cause a more open U37 structure that could potentially increase binding to the gK/UL20 heterodimer, thus, enhancing infectious virus production.

### 4.2. VC2 and FC819S Upregulate GM-CSF Protein Secretion in Different Cell Types

GM-CSF plays a vital role in monocyte, macrophage, and dendritic cell maturation and is critical for antigen presentation. For this reason, the oncolytic HSV-1 virus Talimogene laherparepvec (T-VEC; Imlygic™, Amgen Inc.) was engineered to express GM-CSF to enhance anti-tumor immune responses [50]. We have shown that VC2 is efficacious in treating melanomas in a mouse model and could potentially be utilized as a vector expressing cancer neoantigens to produce a personalized cancer vaccine [27,28]. We observed a significant upregulation of GM-CSF secretion in various cells, including tumor-derived cells infected with VC2. The pattern of GM-CSF secretion was consistent, suggesting that the induction method is consistent across cell lineages and is likely not specific to transformed cells or epithelial cells but is likely found in most cell types. This enhanced GM-CSF secretion may be partially responsible for the demonstrated efficacy of the VC2 virus for melanoma in mice [27,28], as well as for breast cancer in mice (Nabi and Kousoulas, in preparation).

GM-CSF is not highly conserved across species, with humans and mice only sharing 56% homology. This species-specificity necessitates the creation of human or mouse species-specific versions of GM-CSF-inserted viruses. However, because FC819S, VC2, and VC2C819S can stimulate mouse GM-CSF as effectively as human GM-CSF, the viruses can be used across various models without the need for specialized gene tailoring. These qualities lend greater versatility to these models and their broader applications.

We have previously predicted the structure of gK and UL20 using domain-specific modeling. In this manuscript, we utilized newer predictive algorithms to derive the predicted structure of the gK and UL20 heterodimer formation. Notably, the prediction of the gK/UL20 heterodimer having the gK and UL20 amino terminal deletions engineered in the VC2 virus in juxtaposition to their association with cellular membranes reveals that the gK/UL20 heterodimer is shifted inwards into the cellular membranes, potentially altering the availability of cytoplasmic domains of gK/UL20 to bind to other viral and cellular proteins. Notably, the horizontal alpha helix of gK that is shifted from the intracellular space into the plasma membrane is the most highly conserved region of gK, suggesting that it forms a critical function [51]. Additionally, gK and UL20 are important modulators of gB, the fusion protein. Entry requires four essential glycoproteins gB, gD, gH, and gL. Alterations in the gK/UL20 conformational changes to gB likely impact that entire complex (Figure 6A). The introduction of the C819S mutation alters the UL37 overall structure revealing a more open conformation. This potential structure change may allow improved binding of UL37 to the gK/UL20 complex, which is retracted inwards into the plasma membrane, thus, restoring VC2 defects in virus replication and spread.

The observed enhanced secretion of GM-CSF by VC2 and FC819S viruses suggests that the UL37/gK/UL20 complex functions not only in virion morphogenesis and egress but also in the release of GM-CSF into extracellular spaces. Interestingly, VC2 does not replicate efficiently in cGAS-null cell lines, and the UL37C819S mutation restores replication. This result suggests that the UL37/gK/UL20 protein complex has additional functions in immunomodulation, partially by suppressing GM-CSF secretion from infected cells.

## Figures and Tables

**Figure 1 viruses-15-00367-f001:**
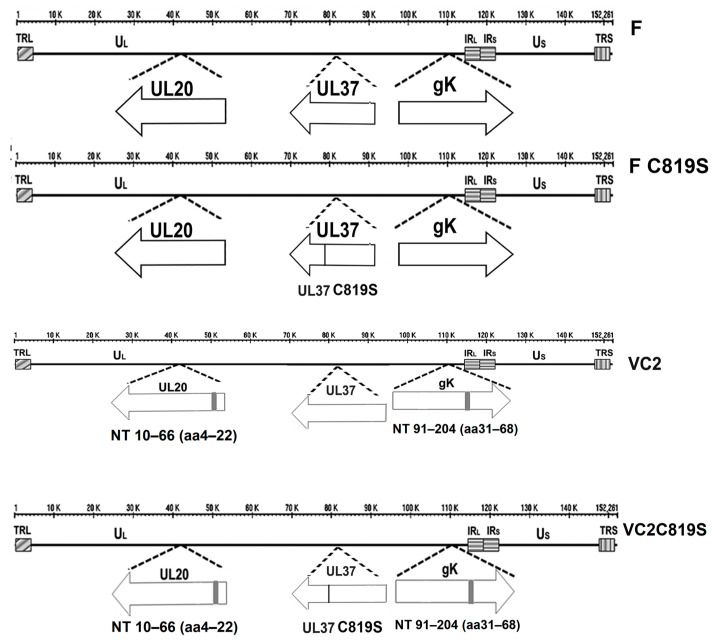
Recombinant virus construction. Schematic of the prototypic arrangement of the HSV-1 genome with expanded regions showing UL20, UL37, and gK genes with deletions marked in black for F (**top**), FC819S (**second**), VC2 (**third**), and VC2C819S (**bottom**).

**Figure 2 viruses-15-00367-f002:**
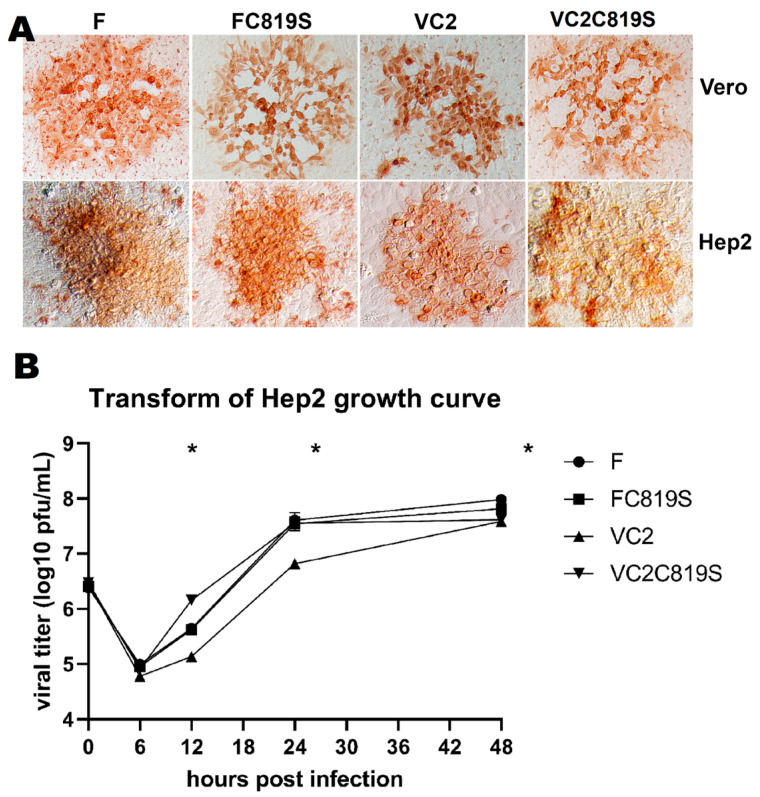
Replication kinetics and plaque morphologies. (**A**) Plaque morphology of F (**left**), FC819S (**second**), VC2 (**third**), and VC2C819S (**right**) 48 h post-infection on Vero (**top**) and Hep2 (**bottom**) visualized by IHC and developed with NovaRed substrate. (**B**) Growth curve of F, FC819S, VC2, and VC2C819S at an MOI of 1.0 on Vero. Whole lysates were collected at 0, 6, 12, 24, and 48 hpi and titrated on Vero. * *p* < 0.05 by one-way ANOVA at each individual time point.

**Figure 3 viruses-15-00367-f003:**
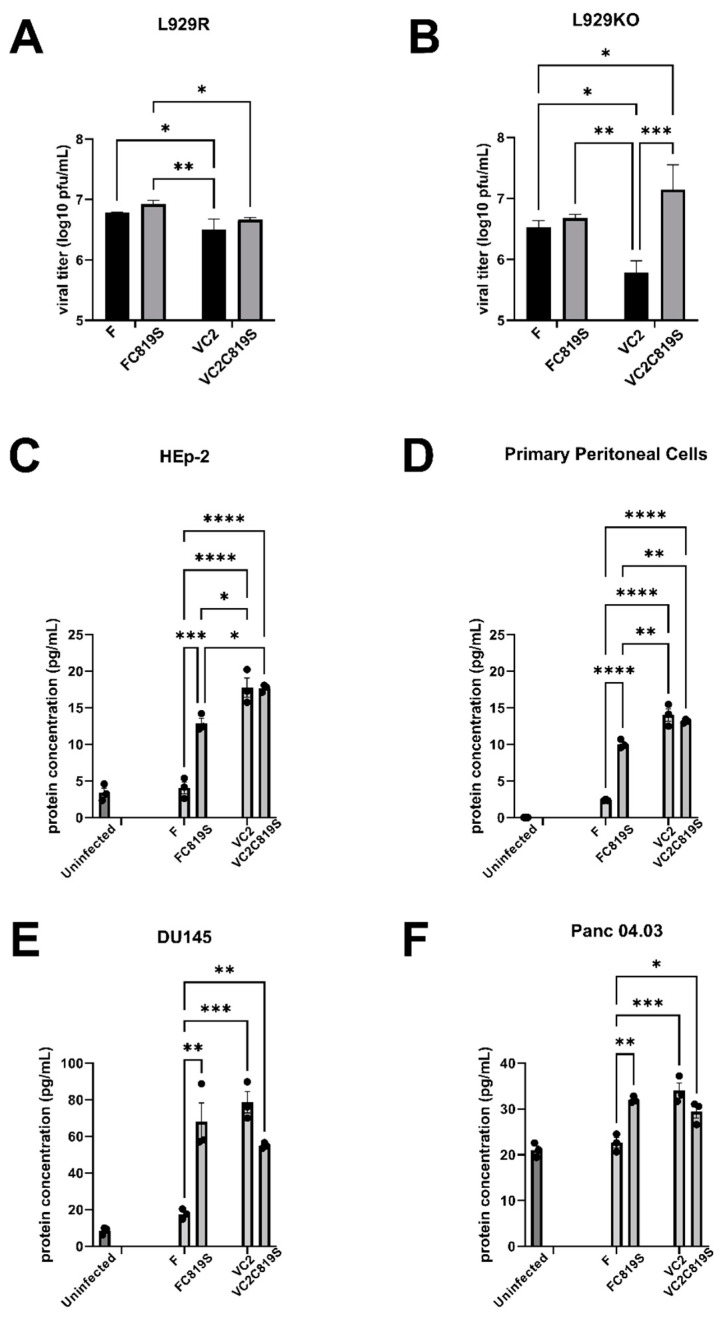
Replication of viruses in the presence of cGAS and expression of GM-CSF. Growth curve of F, FC819S, VC2, and VC2C819S at an MOI of 1.0 on L929R cells (cells reconstituted with human cGAS) (**A**) and L929KO cells (cells with cGAS−/−) (**B**). Whole lysates were collected at 24 hpi and titrated on Vero. Protein expression was measured via ELISA using the conditioned media collected from infected Hep2 cells (**C**), infected primary mouse peritoneal cells (**D**), DU145 (**E**), and Panc 04.03 (**F**) at 12 h post-infection. Statistics were performed via two-way ANOVA of F, FC819S, VC2, and VC2C819S. * *p* < 0.05. ** *p* < 0.01. *** *p* < 0.001. **** *p* < 0.0001.

**Figure 4 viruses-15-00367-f004:**
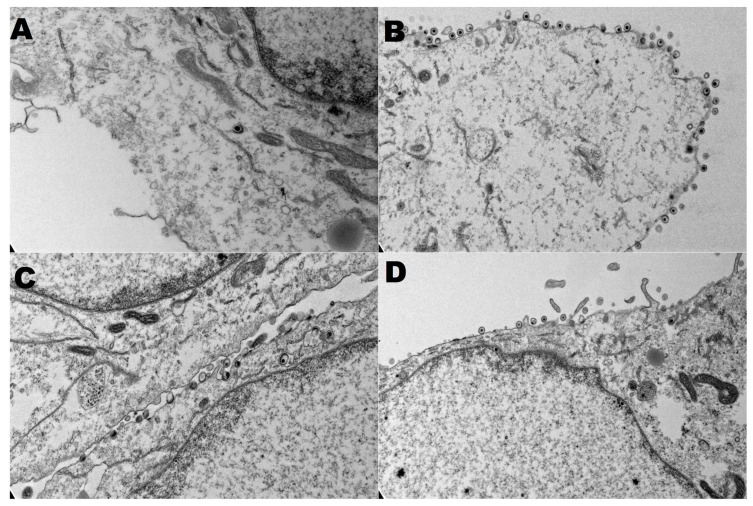
Electron micrographs of wild-type and recombinant viruses. Electron microscopy data of virions at the cellular membranes at 18 h post-infection with F(**A**), FC819S (**B**), VC2 (**C**), and VC2C819S (**D**) at an MOI of 5.

**Figure 5 viruses-15-00367-f005:**
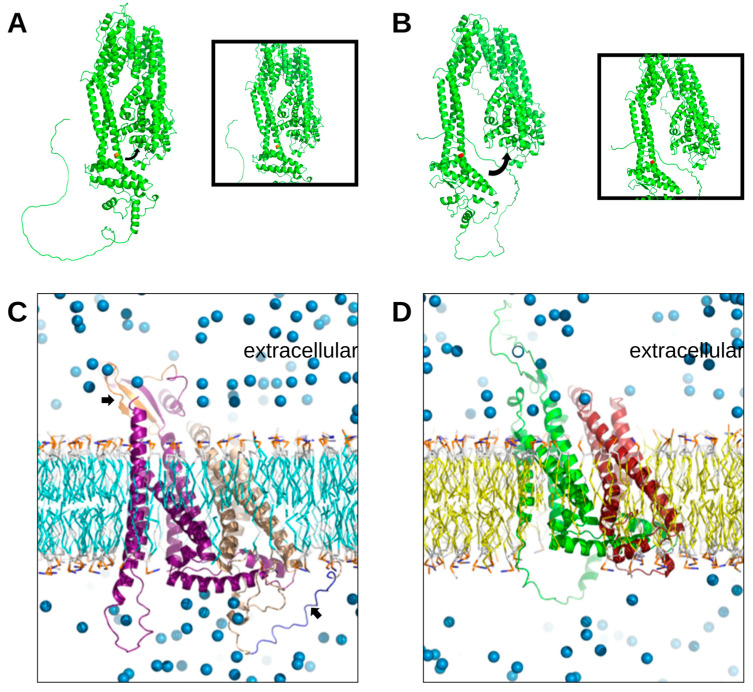
Predicted structures of UL37 protein and gK/UL20 protein complex. The predicted structure of wild-type UL37 (**A**) and UL37 C819S (**B**) were generated using AlphaFold2 with C819 marked in orange and C819S in red. (**C**) Predicted wild-type gK/UL20 complex structure using AlphaFold2-multimer, embedded in a bilipid membrane oriented with the extracellular space above the membrane and the cytoplasmic space above the membrane using MemProtMD. gK is shown in purple, with the 31–68 region in orange in the extracellular space. UL20 is gold, with the 4–22 region in blue in the cytoplasmic space. (**D**) Prediction of gKΔ31–68/UL20Δ4–22 complex structure using AlphaFold2-multimer, embedded in the bilipid membrane where gK is green, and UL20 is red.

**Figure 6 viruses-15-00367-f006:**
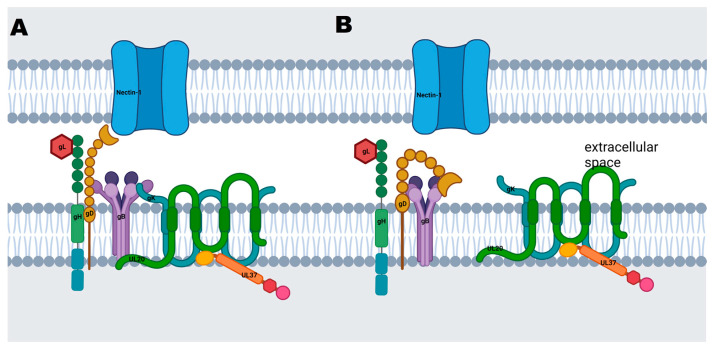
Model of the effect of gK/UL20 deletions on the HSV-1 fusion protein complex. Proposed model of the gK, UL20, and UL37 complex binding with the gH/gL, gD, and gB fusion complex (**A**). Proposed dissociation between the fusion complex and the gKΔ31–68/UL20Δ4–22 complex in the VC2 vaccine (**B**). Schematic was created with biorender.com.

## Data Availability

PDB files of predicted structures are available to the corresponding author upon request. All other data are available within the article.

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
