# Peer review of "Inactivation of the UL37 Deamidase Enhances Virus Replication and Spread of the HSV-1(VC2) Oncolytic Vaccine Strain and Secretion of GM-CSF"

_viruses, 2023, doi:10.3390/v15020367_

Round 1

Reviewer 1 Report

This manuscript by Clark et al. reports the function of UL37 deamidase activity in the context of gK/UL20 complex, as a continued interest of the Kousoulas laboratory. The authors found that the UL37 deamidase activity, similar to the mutations in gK/UL20 of VC2 strain, is important for GM-CSF production. Interestingly, the deamidase-inactivating C819S mutation was able to restore VC2 replication in cultured cells, although it did not further increase GM-CSF production. The authors also present interesting phenotype that HSV-1 UL37C819S mutant (F strain background) was found attached to the plasma membrane by EM studies, a similar phenotype is observed with VC2 strain. To explain their findings, the authors used computer modeling (alpha-fold) to predict the structure of gK/UL20 and UL37. Their observation suggests that the C819S mutation may enable UL37 to restore gK/UL20 association with lipid membrane via pulling the gK/UL20 out from a conformation that is deeply embedded in the membrane. These findings are informative and interesting to the field, and provide further direction for future investigations. The following points the author may consider to address:

Main points:

1.     Most of the cytokine (GM-CSF) and viral replication data shown in Figure 3 were presented at single time point. It is not clear whether the authors have data for multiple time points but only one was presented. This needs some clarification. An important point related to this that, when viral titer is considered, the HSV-1 yield in L929KO cells was not higher than that in L929R cells. Perhaps, validating cGAS expression and its downstream signaling is necessary. 

2.     Data presented in Figure 2A may not show the significant difference in plaque size and CPE. A picture of lower magnification with multiple plaques can be presented in replace of these single plaque view. Additional quantification of plaque size and distribution of various plaque size may provide more convincing results. 

3.     A remarkable phenotype of UL37C819S mutant was presented in Figure 4B, compared to its wildtype virus. However, when viral titer was determined in the extracellular and intracellular compartment, there was no difference. Any comments on these observations?

Minor points

1.     The manuscript is generally well written. We noted several minor spelling or glitches.

2.     Line 382, it should be “without” the need for specialized gene tailoring?

3.     Line 409: UL37/gK/UL20.

Author Response

Reviewer 1:

Main points:

Comment 1.    Most of the cytokine (GM-CSF) and viral replication data shown in Figure 3 were presented at single time point. It is not clear whether the authors have data for multiple time points but only one was presented. This needs some clarification. An important point related to this that, when viral titer is considered, the HSV-1 yield in L929KO cells was not higher than that in L929R cells. Perhaps, validating cGAS expression and its downstream signaling is necessary. 

Response A:   Multiple time points were studied in HEp-2 cells. However, only the 12 hours post-infection (hpi) data point was presented in this paper. GM-CSF is stable in cell culture media, so secreted GM-CSF accumulates in the media over time, demonstrating a cumulative effect rather than a transitory state. At earlier points (<12hpi) GM-CSF increased secretion is observed at low levels achieving significance at 12 hpi.

Response B: The L929KO and L929R cell lines were stably transduced and validated in the published paper  (Zhang, J., Zhao, J., Xu, S., Li, J., He, S., Zeng, Y., Xie, L., Xie, N., Liu, T., Lee, K., Seo, G.J., Chen, L., Stabell, A.C., Xia, Z., Sawyer, S.L., Jung, J., Huang, C., Feng, P., 2018. Species-Specific Deamidation of cGAS by Herpes Simplex Virus UL37 Protein Facilitates Viral Replication. Cell Host Microbe 24, 234-248 e235). Specifically, these authors reported no significant difference in infectious virus production in in the L929KO compared to L929R cells.

Our results are consistent with their published data.

Comment 2:     Data presented in Figure 2A may not show the significant difference in plaque size and CPE. A picture of lower magnification with multiple plaques can be presented in replace of these single plaque view. Additional quantification of plaque size and distribution of various plaque size may provide more convincing results. 

Response:      In response to this comment, we have now added supplemental figure s3 showing the entire 12-well plate showing many viral plaques.

Comment 3:     A remarkable phenotype of UL37C819S mutant was presented in Figure 4B, compared to its wildtype virus. However, when viral titer was determined in the extracellular and intracellular compartment, there was no difference. Any comments on these observations?

Response:      We consider that the resultant virus production differences are most likely due to overall viral replication efficiency rather than decreased egress from infected cells.

 Minor points

Comment 1:    The manuscript is generally well written. We noted several minor spelling or glitches.

Response:      We have re-examined the manuscript and corrected few typos and other errors.

Comment 2:    Line 382, it should be “without” the need for specialized gene tailoring?

Response:      This suggestion has been rectified.

Comment 3:    Line 409: The UL37/gK/UL20 designation has been corrected as suggested.

Reviewer 2 Report

Clark and colleagues expand work on a vaccine/oncolytic F strain of HSV-1 that was attenuated by N-terminal deletions of UL20 and gK. These are envelope proteins known to be involved in syncytia and impact the function of gB. These viruses are safe in as much as they fail to enter sensory neurons but continue to replicate in epithelial cells. However replication was compromised to some extent and virus yield and spread (reduced plaque size) were partially impaired. The vector also produced GMCSF to encourage antigen presenting cell recruitment to infected tumors for example. Oncolytic HSVs are likely to function better if they replicate and spread efficiently. They queried whether mutational inactivation of the Ul37 deamidase domain by a single amino acid substitution at residue C819S would impact the efficiency of virus replication in the context of WT HSV-1(F) or UL20/gK deletion mutants.  UL37 is a tegument protein that functions together with UL20 and gK to achieve virus envelopment and egress. UL37 also can deamidate cGAS and RIG-I to reduce anti-viral innate responses to infection. They made the interesting observation that the UL37 mutant in combination with deltaUL20/gK rescued virus replication and enhanced plaque size compared with the double deletion mutant suggesting the oncolytic function would be improved. This was not tested but is apparently under investigation. The UL37 mutation did not impact the growth or plaque size of the parental F strain virus but essentially restored WT virus replication in VC2, the double mutant. The mechanism remains obscure despite speculative protein modeling. This is a nicely written paper although the Intro contains a lot of somewhat irrelevant info. The logic behind rescue of the growth of VC2 is clear but there are easier ways to attenuate the F strain that do not compromise replication in tumor cells and do not affect innate immunity. Have the authors compared the standard ICP34.5 mutants with their VC2 vector.

Comments:

1.     A key piece of data that could explain the mechanism may have to do with the impact of the Ul37 mutation on association with the mutant UL2/gK products. Is deamidation involved in the envelopment/egress process? Apparently, its role in cGAS inactivation is not relevant to the double mutant rescue.

2.     The impairment of the ability of mutant UL37 to knock down cGAS/RIG-I activity reduced the ability to replicate since it might be anticipated that there would be an increase in type 1 IFN production. Is there a change in IRGs in the different conditions (mutant combinations) of infection. VC2C819S replication was slightly less than the WT virus and the FC819S mutant in cGAS+ cells however KO of cGAS significantly increased the rescued VC2 mutant. It would seem that compromising the UL37 function may not be helpful to virus replication in tumors. Tumors often have IFNa defects, so IRG activity becomes relevant to understanding the importance of the UL37 mutation. Have the authors assessed replication in tumor cells with known cGAS status?

3.     The underlying mechanism by which the UL37 deamidation mutant rescues the double mutant is not addressed biochemically. The authors make an attempt to describe the mechanism using protein modeling in membranes. The hypothesis is that a predicted change in the structure of the UL37 mutant somehow influences the function of the double deletion of Ul20/gK. Does the UL37 mutant show a change in its association with the mutant 20/K complex. The link is unclear. Since the UL37 mutant makes enveloped virus, its deamidation function seems to be uncoupled from envelopment.  Do the WT and mutant UL37 products differentially interact with the mutated 20/K products.

4.     The secretion of GMCSF was increased in the rescued VC2 vector presumably due to enhanced virus replication. Was replication assessed in cells with increased GMCSF production. The authors suggest that GMCSF secretion is enhanced however there are no experiments to query this.

Author Response

Reviewer 2:

Comment 1:    A key piece of data that could explain the mechanism may have to do with the impact of the Ul37 mutation on association with the mutant UL20/gK products. Is deamidation involved in the envelopment/egress process? Apparently, its role in cGAS inactivation is not relevant to the double mutant rescue.

Response:      The authors agree with the reviewer that the UL37/cGAS interaction is likely not relevant to the double mutant rescue. We would like to direct the reviewer to Figure 4, which shows the electron microscopy photos of the egress differences between F, FC819S, VC2, and VC2C819S. Notable, the UL37C819S mutants appear to cling to the cellular membrane during egress, which is not observable in HSV-1(F) or VC2. This may indicate an altered envelopment/egress process as the reviewer suggests.

Comment 2:    The impairment of the ability of mutant UL37 to knock down cGAS/RIG-I activity reduced the ability to replicate since it might be anticipated that there would be an increase in type 1 IFN production. Is there a change in IRGs in the different conditions (mutant combinations) of infection. VC2C819S replication was slightly less than the WT virus and the FC819S mutant in cGAS+ cells however KO of cGAS significantly increased the rescued VC2 mutant. It would seem that compromising the UL37 function may not be helpful to virus replication in tumors. Tumors often have IFNa defects, so IRG activity becomes relevant to understanding the importance of the UL37 mutation. Have the authors assessed replication in tumor cells with known cGAS status?

Response:      The viruses have been grown in Vero cells which are IFNa and IFNb deficient and express very low amounts of cGAS or STING. The viruses exhibit similar growth deficiencies in Vero cells as they have in HEp-2 cells. Additionally, the DU145 cells (prostate cancer cells) used in Figure 3 have been utilized as a tumor model for cGAS/STING suppression (Suter, M.A., Tan, N.Y., Thiam, C.H., Khatoo, M., MacAry, P.A., Angeli, V., Gasser, S., Zhang, Y.L., 2021. cGAS-STING cytosolic DNA sensing pathway is suppressed by JAK2-STAT3 in tumor cells. Sci Rep 11, 7243). Notably, these viruses were able to grow efficiently in those cells.

Comment 3:    The underlying mechanism by which the UL37 deamidation mutant rescues the double mutant is not addressed biochemically. The authors make an attempt to describe the mechanism using protein modeling in membranes. The hypothesis is that a predicted change in the structure of the UL37 mutant somehow influences the function of the double deletion of Ul20/gK. Does the UL37 mutant show a change in its association with the mutant 20/K complex. The link is unclear. Since the UL37 mutant makes enveloped virus, its deamidation function seems to be uncoupled from envelopment.  Do the WT and mutant UL37 products differentially interact with the mutated 20/K products.

Response:      We agree with the reviewer that one potential explanation of the observed phenotype is a potential alteration of protein-protein interactions between UL37 and the gK/UL20 heterodimer.  However, these protein-protein interactions are very difficult to assess via immunoprecipitation experiments due to the high hydrophobic nature of the gK/UL20 heterodimer. Furthermore, modeling of the entire UL37/gK/UL20 complex was not possible with AlphaFold 2 Multimer software due to the UL37 size.  An alternative hypothesis is that both UL37 and gK/UL20 affect similar aspects of intracellular virus replication. It is likely that the deamidation site of UL37 has other deamidation targets that are currently unknown, and interactions and signaling through these proteins result in the observed changes

Comment 4:    The secretion of GMCSF was increased in the rescued VC2 vector presumably due to enhanced virus replication. Was replication assessed in cells with increased GMCSF production. The authors suggest that GMCSF secretion is enhanced however there are no experiments to query this.

Response:      We respectfully disagree with the reviewer’s interpretation of the presented data. Figure 3C shows that infection with wild-type HSV-1(F) does not increase GM-CSF secretion relative to uninfected cells. Additionally, Figure 2b shows that VC2 has the lowest replication in HEp-2 cells. However, Figure 3C shows that VC2 also has the highest secretion of GM-CSF in HEp-2 cells. This suggests that GM-CSF secretion is not directly related to the level of viral replication and is more likely dependent on signaling events.